# Phage Therapy in the Era of Multidrug Resistance in Bacteria: A Systematic Review

**DOI:** 10.3390/ijms23094577

**Published:** 2022-04-21

**Authors:** Carlos Aranaga, Lady Daniela Pantoja, Edgar Andrés Martínez, Aura Falco

**Affiliations:** 1Chemistry and Biotechnology Research Group (QUIBIO), School of Basic Sciences, Universidad Santiago de Cali, Pampalinda Campus Calle 5 # 62-00, Santiago de Cali 76001, Colombia; lady.pantoja01@usc.edu.co (L.D.P.); edgar.martinez00@usc.edu.co (E.A.M.); 2Microbiology, Industry and Environment Research Group (GIMIA), School of Basic Sciences, Universidad Santiago de Cali, Pampalinda Campus Calle 5 # 62-00, Santiago de Cali 76001, Colombia; aura.falco00@usc.edu.co

**Keywords:** phage therapy, bacteriophages, multidrug-resistant bacteria, infections

## Abstract

Bacteriophages offer an alternative for the treatment of multidrug-resistant bacterial diseases as their mechanism of action differs from that of antibiotics. However, their application in the clinical field is limited to specific cases of patients with few or no other alternative therapies. This systematic review assesses the effectiveness and safety of phage therapy against multidrug-resistant bacteria through the evaluation of studies published over the past decade. To that end, a bibliographic search was carried out in the PubMed, Science Direct, and Google Scholar databases. Of the 1500 studies found, 27 met the inclusion criteria, with a total of 165 treated patients. Treatment effectiveness, defined as the reduction in or elimination of the bacterial load, was 85%. Except for two patients who died from causes unrelated to phage therapy, no serious adverse events were reported. This shows that phage therapy could be an alternative treatment for patients with infections associated with multidrug-resistant bacteria. However, owing to the phage specificity required for the treatment of various bacterial strains, this therapy must be personalized in terms of bacteriophage type, route of administration, and dosage.

## 1. Introduction

Bacteriophages were discovered by Frederick Twort and Félix d’Hérelle in 1915 and 1917, respectively. Since then, it has been suggested that the administration of these viruses could be used to treat bacterial infections. Initial results with phage therapy (PT) were promising; however, its application was limited owing to the discovery of penicillin in 1928 as well as the large-scale production of antibiotics in 1940 [1,2,3,4]. In recent years, the use of PT has recommenced owing to the increased bacterial resistance to antibiotics, which can arise because of mutations or be acquired by the acquisition of resistance-conferring genes via horizontal gene transfer (HGT). HGT may occur via conjugation, transformation, and transduction processes as well as via mobile genetic elements, such as transposons, and insertion sequences [5,6].

Infections caused by multidrug-resistant bacteria have become one of the leading causes of morbidity and mortality worldwide, with approximately 700,000 deaths each year [7,8]. The World Health Organization (WHO) has listed critical priority pathogens that require new antibiotics or therapeutic alternatives; these pathogens include multidrug-resistant bacteria that are especially dangerous in hospitals and nursing homes and for patients who need to be treated with ventilators and intravenous catheters [9]. According to the Organization for Economic Cooperation and Development (OECD), in Europe, North America, and Australia alone, 2.4 million people could die between now and 2050 if current bacterial resistance rates continue [10], which evidences that bacterial resistance poses a global public health problem [2,3,4,5,6,11]. Likewise, the discovery and development of new antibiotics have decreased because of the large amount of time and money required, resulting in increasingly difficult clinical management of infections and, in some cases, infections that are impossible to treat [9,11,12]. For these reasons, there is a need to look for alternatives for patients for whom conventional antibiotic therapy is ineffective. Thus, PT has become an alternative as its mechanism of action differs from that of antibiotics [13].

Lytic bacteriophages have had an impact on the clinical treatment of multidrug-resistant bacteria because of their capacity to naturally control bacterial populations [14,15]. These viruses provide novel advantages, such as the safe treatment of infections, as they are harmless to eukaryotic cells, do not cause harmful side effects, and demonstrate high host specificity. This is because they only replicate in the presence of bacteria causing the infection, thus reducing damage to the natural microflora. In addition, genetic exchange between phages rarely happens [16,17,18]. Therefore, as the frequency of therapy increases, the treatment doses and periods needed to achieve an optimal effect are expected to decrease [16,17]. Phage therapy has been used for the treatment of infections related to burn injuries or soft tissue and skin trauma, osteomyelitis, sepsis, bacteremia, and otitis media as well as urinary tract, pulmonary, and prosthetic device-associated infections. It is often used after the generally prolonged stay of the patients, who have been treated with different antibiotics without being able to eradicate the infection successfully [19,20]. However, effectiveness- and safety-related data for this type of therapy are scattered across individual clinical case reports owing to the few clinical studies conducted. In this review, we summarize the effectiveness- and safety-related data from studies conducted on human patients infected with various types of bacteria who underwent PT as an adjuvant or alternative therapy to antibiotics.

## 2. Materials and Methods

### 2.1. Search Strategy

To carry out the bibliographic search, three databases were selected: ScienceDirect, PubMed, and Google Scholar. The keywords used were phage therapy and multidrug resistant. The search was defined with the use of Boolean operators as follows: “phage therapy” AND “multidrug resistant”. The search was adjusted in each database according to the descriptors and using “phage therapy in the era of multidrug resistance in bacteria” as a reference. The search protocol was conducted by two independent reviewers.

### 2.2. Inclusion and Exclusion Criteria

The review included full-text articles and clinical case reports published in Spanish and English over the last ten years, that is, from 1 January 2011 to 25 April 2021. The main objectives of these studies were to describe the use of PT in patients infected with various bacteria, provided that, at the end of the study, it would be possible to tell whether there was a positive or negative response to the treatment.

The review also excluded bibliographic material that had not been reviewed by academic peers; articles written in a language other than Spanish or English; articles published outside the selected duration; bibliographic material, such as reviews, systematic reviews, posters, conferences, book sections, and perspectives; multidrug-resistance-based studies that did not use phage therapy; studies on metagenomics in bacteriophages without subsequent application in human patients; studies on PT in animals; and studies on the use of phages in foods.

The selection of articles was carried out considering the PRISMA statement (Preferred Reporting Items for Systematic Reviews and Meta-analyses) (Figure 1).

### 2.3. Data Extraction

The citations selected, along with their title, were imported into *EndNote*, a Bibliographic Reference Management Software, and Microsoft Office Excel, which was used to check duplicated results between the three databases. Duplicated records were eliminated.

### 2.4. Data Analysis

Evaluation of the records was carried out by reading the titles and the abstracts. All those that did not meet the inclusion criteria were discarded. Then, the full texts were analyzed to rule out those articles that did not meet the inclusion criteria, thus leaving us with the final records that were included in this review. Subsequently, the information of each article was collected in two Excel tables, whereas the most relevant information was captured in graphs for later analysis.

## 3. Results

The bibliographic search yielded a total of 1500 articles distributed as follows: 203 in PubMed, 306 in ScienceDirect, and 991 in Google Scholar published between 2011 and 2021. After excluding duplicated records (336 articles) and 1121 articles that did not meet the inclusion criteria, 27 articles were included in this review (Figure 1). Subsequently, information concerning the effectiveness and safety of PT was collected along with other findings that the authors considered relevant [21].

Of a total of 165 patients who underwent phage therapy, 141 (85%) showed a reduction in or complete elimination of the bacterial load accompanied by an improvement in the signs and symptoms, whereas PT was ineffective in the remaining 24 (15%). Only 21% (*n* = 35) of patients received PT combined with antibiotics, with a 100% success rate. The remaining 79% (*n* = 130) were treated exclusively with PT, with a success rate of 81%. Bacterial resistance to phages was reported in 6 of the 27 articles reviewed [22,23,24,25,26,27]; in four studies, the initial therapy was modified by changing or including new bacteriophages [22,23,24,25,26,27], whereas in the remaining studies, resistance was measured at the end of the study.

The age of the patients receiving PT ranged from 2 to 88 years, with an average of 54 years. Three of these studies were conducted on minors (12%) [19,28,29], whereas 88% included patients over the age of 18. Most cases yielded positive results in terms of effectiveness (improving the medical condition), and adverse side effects were minimal. In addition, phage administration was not associated with the death of any of the patients (Table 1).

A total of 61 multidrug-resistant bacterial species were reported, including *Staphylococcus aureus* (24.5%, *n* = 15/61), *Pseudomonas aeruginosa* (22.9%, *n* = 14/61), *Klebsiella pneumoniae* (13.1%, *n* = 8/61), and *Acinetobacter baumannii* (8.1%, *n* = 5/61) (Figure 2).

In most clinical cases, phage cocktails were administered (84%, *n* = 21), i.e., a combination of 2 to 12 bacteriophages, whereas in seven cases, a single phage (16%) was administered [23,26,30,31,32]. The routes of administration of the phage cocktails in these 27 cases were as follows: 4 topical (16%) [23,33,34,35], 7 intravenous (28%) [19,25,30,33,34,35,36], 4 in organs or cavities (16%) [22,24,32,37,38], 5 through inhalation (16%) [26,27,29,39], and 6 using more than one route of administration (24%) [28,31,40,41,42,43] (Table 2).

**Table 1 ijms-23-04577-t001:** Data collection, after reading the full texts of the selected articles.

Author(s)/ Reference	Country	Year of Publication	Multidrug-Resistant Bacteria	Site or Type of Infection	Patients (Number/Gender/ Age)	Comorbidities/ Medical History	Adverse Effects	Effect of the Treatment
Ooi, M.L. et al. [37]	Australia	2019	*Staphylococcus aureus*	Recalcitrant chronic rhinosinusitis caused by a biofilm	9/4 men, 5 women/ 18–70 years of age	Endoscopic sinus surgery	Diarrhea, epistaxis, oropharyngeal pain, cough, rhinalgia, and low blood bicarbonate level	Positive treatment effect: 2 patients were cured and the others showed reduced bacterial growth
Schooley, R.T. et al. [25]	U.S.	2017	*Acinetobacter baumannii*	Pancreatic pseudocyst infection	1/male/68 years of age	Diabetic, necrotizing pancreatitis	Not reported	Positive treatment effect: patient showed continuous clinical improvement, woke up from coma, showed renal improvement, and was extubated
Jault, P. et al. [23]	France–Belgium	2018	*Pseudomonas aeruginosa- Escherichia coli*	Infection of wounds from burns on the skin	12/men and women/ over 18 years of age	Hospitalized patients, treated for burns in a burn unit	No adverse effects related to PT were observed	No/insufficient treatment effect owing to the administration of a very low treatment dose
Law, N. et al. [34]	U.S.	2019	*Pseudomonas aeruginosa*	Respiratory tract infection	1/woman/26 years of age	Patient with cystic fibrosis, persistent respiratory failure, and renal failure	No adverse effects related to PT were observed	Positive treatment effect: pneumonia clinically resolved, baseline renal function was restored, and fever disappeared
Aslam, S. et al. [19]	U.S.	2019	*Pseudomonas aeruginosa-Burkholderia dolosa-Mycobacterium abscessus*	Lower respiratory tract infections	3/1 man, 2 women/ 67, 28, and 57 years of age	Lung transplantation and progressive kidney failure	Intravenous and nebulized formulations of bacteriophages were well tolerated without adverse effects attributed to bacteriophage administration	A positive effect of the treatment was observed in two of the three patients (one male and female, each aged 67 years) as they did not develop further infection with *Pseudomonas aeruginosa*, and their respiratory status markedly improved. The 28-year-old patient died. However, this was not attributed to PT, but rather to a sequence of events that led to multi-organ failure, most plausibly attributable to postsplenectomy complications and liver disease. Hence, a verdict on the treatment was not possible.
Nir-Paz, R. et al. [35]	U.S.	2019	*Acinetobacter baumannii-Klebsiella pneumoniae*	Left tibial infection following trauma	1/male/42 years of age	Grade IIIA bilateral open fractures of the lower extremities: left bicondylar tibial plateau fracture with compartment syndrome and right distal femoral fracture	No adverse effects associated with PT were observed	Positive treatment effect, with healing of the graft and elimination of subtle chronic bone pain in the left leg. During a follow-up period of 8 months after treatment, no Ab- or Kp-positive cultures were obtained from any site.
Tkhilaisshvili, T. et al. [33]	Germany	2019	*Pseudomonas aeruginosa- Staphylococcus epidermidis*	Chronic recurrent periprosthetic infection of the knee and chronic osteomyelitis of the femur	1/woman/ 80 years of age	Metabolic syndrome (type II diabetes mellitus and obesity) and chronic renal failure	No adverse effects related to PT were observed	Positive treatment effect. The patient did not report pain in the right knee; the soft tissue at the surgical site did not show complications; and mobility was satisfactory. All periprosthetic tissue samples collected intraoperatively resulted in negative cultures.
Corbellino, M. et al. [28]	Italy	2019	*Klebsiella pneumoniae*	Infections of the gastrointestinal and urinary tracts	1/male/57 years of age	Crohn’s disease, obstructive nephrolithiasis, right nephrectomy and radical cystectomy, and stage III chronic renal failure	No adverse effects related to PT were observed	Positive treatment effect. Long-lasting multi-site colonization by an MDR Kp strain in a patient with a single kidney, a cutaneous ureterostomy, and an indwelling ureteral stent resolved after a 3-week course of PT.
Duplessis, C. et al. [36]	U.S.	2017	*Pseudomonas aeruginosa*	Recalcitrant infection after ASD/VSD closures	1/male (child)/2 years of age	DiGeorge syndrome and complex congenital heart disease, including a type B interrupted aortic arch, posterior misalignment of a ventricular septal defect (VSD), sub-aortic stenosis, bicuspid aortic valve, and secundum atrial septal defect	Decompensation owing to anaphylaxis, which was subsequently attributed to progressive heart failure, although the release of endotoxins as a contributing factor could not be ruled out	Clinical improvement was seen after treatment for several days with PT; however, the patient decompensated and developed severe arrhythmias and cardiac and septic shocks. This turbulent worsening was attributed to the progressive accumulation of undrained fluids, a history of influenza infection, and end-stage heart failure. Finally, the child passed away.
Qin, J. et al. [24]	China	2021	*Klebsiella pneumoniae*	Multifocal urinary tract infections	1/male/66 years of age	The patient’s cancerous bladder was partially resected	No adverse effects related to PT were observed	Negative treatment effect. The phage cocktail did not work. It is believed that the heterogeneous bacteria that colonized the renal pelvis could not be effectively eliminated because the phages could not reach them.
Tan, X. et al. [26]	China	2021	*Acinetobacter baumannii*	Lung infection	1/male/86 years of age	Exacerbation of chronic obstructive pulmonary disease and type II diabetes	No adverse effects related to PT were observed	Positive treatment effect, with elimination of the pathogen and clinical improvement of the patient’s lung function
Hoyle, N. et al. [29]	Georgia	2018	*Achromobacter xylosoxidans*	Chronic lung infection	1/female/ 17 years of age	Cystic fibrosis	No adverse effects related to PT were observed	Positive treatment effect; dyspnea resolved and cough was reduced. Lung function: FEV1 increased.
Leitner, L. et al. [38]	Georgia	2018	*Enterococcus* spp. *Escherichia coli*, *Streptococcus* spp.	Urinary tract infection	28/men/ over 18 years of age	Transurethral resection of the prostate (TURP), with complicated or recurrent uncomplicated UTI without signs of systemic infection	No adverse effects related to PT were observed	Positive effect of treatment in 60% of patients, with normalization of urine cultures
Cano, E.J. et al. [30]	U.S.	2020	*Klebsiella pneumoniae*	Knee joint prosthesis infection	1/male/62 years of age	Diabetes mellitus, obesity, total knee arthroplasty, multiple bacterial infections	No adverse effects related to PT were observed	Positive treatment effect. The patient showed improvement and remained asymptomatic 34 weeks after completing the treatment, in addition to showing improvement of erythema, swelling, pain, range of motion, and function of his right lower extremity.
Gupta, P. et al. [44]	Varanasi, India	2019	*Escherichia coli*, *Staphylococcus aureus*, and *Pseudomonas aeruginosa*	Infection of chronic wounds (ulcers) that do not heal	20/12–60 years of age	Not reported	No adverse effects related to PT were observed	Positive treatment effect in 7 patients who completely healed; the others showed significant improvement in terms of wound healing
Lebeaux, D. et al. [45]	Paris, France	2021	*Achromobacter xylosoxidans*	Persistent lung infection	1/male/12 years of age	Cystic fibrosis and double lung transplantation	No adverse effects related to PT were observed	Positive effect. Respiratory condition gradually improved, and the bacterial load decreased.
Nadareishvili, L. et al. [40]	Georgia	2020	*Staphylococcus aureus*	Osteomyelitis of the sternum and parasternal abscess	1/male/74 years of age	Coronary artery graft, type II diabetes mellitus, high blood pressure	Not reported	Positive treatment effect. The diameter of the wound decreased and the purulent fistula closed.
			*Staphylococcus aureus*	Chronic osteomyelitis of the right tibia	1/female/ 60 years of age	Type II diabetes mellitus	Not reported	Positive treatment effect with complete wound closure
			*Burkholderia cepacia*, *S. aureus*, and *Enterococcus faecalis*	Osteomyelitis with foot ulcer	1/male/69 years of age	Type II diabetes mellitus	Not reported	Positive treatment effect. The patient improved and the ulcer healed.
			*Pseudomonas aeruginosa*, then by *Staphylococcus aureus* and *Serratia Marcescens*	Laryngeal (post-surgical) infection	1/male/68 years of age	Removal of the larynx due to carcinoma	Not reported	Positive treatment effect. The patient improved and infection was completely cured.
Patel, D.R. et al. [46]	India	2019	*Escherichia coli*, *Pseudomonas aeruginosa*, *Staphylococcus aureus*, *Klebsiella pneumoniae*, *Proteus species*, *Citrobacter freundii*, *Morganella morganii*, *Acinetobacter baumannii*	Chronic wound infection of the skin	48/34 men, 14 women/12–70 years of age	27 diabetics, 8 with high blood pressure, 2 with tuberculosis, and 10 with amputations	Not reported	Positive treatment effect in 39 patients who were completely cured; 2 patients died
Fadlallah, A. et al. [31]	Georgia	2015	*Staphylococcus aureus*	Eye infection/bacterial keratitis	1/female/65 years of age	Postoperative corneal abscess of the left eye and interstitial keratitis after craniotomy for acoustic neurinoma	No adverse effects related to PT were observed	Positive treatment effects, with stabilization of ocular signs and negative eye and nasal cultures
Jennes, S. et al. [41]	Belgium	2017	*Pseudomonas aeruginosa*	Acute kidney infection after injury	1/male/61 years of age	Peritonitis owing to *Enterobacter cloacae* and severe abdominal sepsis with disseminated intravascular coagulation secondary to strangulated diaphragmatic hernia. The patient had a prolonged hospital stay complicated by peripheral gangrene of his extremities, which resulted in the amputation of the lower limbs and the development of large necrotic pressure ulcers.	No adverse effects related to PT were observed	Positive treatment effects; blood cultures were negative, CRP levels decreased, and fever disappeared. Kidney function recovered within a few days. However, the patient died after 4 months from a sudden refractory cardiac arrest in the hospital due to *Klebsiella pneumoniae.*
Kuipers, S. et al. [42]	Holland	2020	*Klebsiella pneumoniae*	Recurrent urinary tract infection	1/male/58 years of age	Kidney transplant	No adverse effects related to PT were observed	Positive treatment effects. The symptoms of urethritis quickly disappeared.
Onsea, S. et al. [43]	Belgium	2019	*P. aeruginosa* and *S. epidermidis*	Chronic osteomyelitis of the pelvis	Patient 1	Isolated fibrous tumor in the left pelvic region	Not reported	Positive treatment effects. C-reactive protein and white blood cell levels returned to normal.
			*P. aeruginosa* and *S. epidermidis*	Non-union of distal femur	Patient 2	Polytrauma after aggression, with open segmental fractures of the right femur	Not reported	Positive treatment effects. The levels of C-reactive protein and white blood cells returned to normal. Infection-free.
			*S. agalactiae* and *S. aureus*	Postoperative problems of the femur wound	Patient 3	Polytrauma after the collapse of a building, with crush injuries of the upper right leg and complex fractures of the femur	Not reported	Positive treatment effects. C-reactive protein and white blood cell levels returned to normal.
			*E. faecalis*	Infection of the surgical site with abscess formation and evolution to osteomyelitis of the femur	Patient 4	Polytrauma after traffic accident, with fracture of the femur	Local redness and pain	Positive treatment effects. C-reactive protein and white blood cell levels returned to normal.
Wu, N. et al. [27]	Shanghai, China	2021	*Acinetobacter baumannii*	Respiratory tract infection acquired in the hospital	1/male/62 years of age	Critical condition due to COVID-19 (RNA-negative) and co-infection with *Candida albicans* and *Ralstonia mannitolilytica*	Cytokine storm and fever 4 h after 1Ф phage administration	Positive treatment effects. Clinical improvement, discharged from the hospital on day 30.
				Respiratory tract and intubation wound infections acquired in the hospital	1/male/64 years of age	Critical condition due to COVID-19 (RNA-negative)	No adverse effects related to PT were observed	Positive treatment effects. Clinical improvement and discharge from the hospital on day 9.
				Respiratory tract infection acquired in the hospital	1/male/81 years of age	Critical condition due to COVID-19 (RNA-negative) and co-infection with *Candida albicans* and *Candida glabrata*	No adverse effects related to PT were observed	Positive treatment effects. The infection was eliminated; however, patient died on day 10 owing to respiratory failure caused by infection with *Klebsiella pneumoniae.*
				Respiratory tract and bladder infections acquired in the hospital	1/male/78 years of age	Critical status due to COVID-19 (RNA-negative) and co-infection with *Candida albicans*, *Klebsiella pneumoniae* resistant to carbapenems, and *Sphingomonas paucimobilis*	No adverse effects related to PT were observed	Positive treatment effect with improvement, patient was discharged from the ICU on day 7; however, he died on day 40 owing to respiratory failure
Bao, J. et al. [22]	Shanghai, China	2020	*Klebsiella pneumoniae*	Recurrent urinary tract infection	1/female/63 years of age	Type II diabetes and hypertension	No adverse effects related to PT were observed	Positive treatment effect. The infection was completely eradicated.
Rose, T. et al. [39]	Brussels, Belgium	2014	*Pseudomonas aeruginosa* and *Staphylococcus aureus*	Burn infection	9/4 men, 5 women/ 27–88 years of age	Not reported	No adverse effects related to PT were observed	Negative treatment effect. Low bacterial load, almost unchanged.
Rubalskii, E. et al. [32]	Germany	2020	*Staphylococcus aureus*, *Enterococcus faecium*, *Pseudomonas aeruginosa*, *Klebsiella pneumoniae*, and *Escherichia coli*	Patients had infections associated with immunosuppression after organ transplantation or had infections of vascular grafts, implanted medical devices, and surgical wounds	8/7 men, 1 woman	Not reported	No adverse effects related to PT were observed	Positive effect of treatment in 7 patients who are completely cured; the other showed a significant improvement in terms of decreased bacterial load

## 4. Discussion

This systematic review displays effectiveness- and safety-related data as well as some relevant findings that have been reported in PT studies in humans from studies published in English over the last ten years. Based on the findings from a total of 165 patients treated in 27 selected studies, it can be concluded that PT produced encouraging results for the treatment of infections caused by various bacterial species, especially those that are difficult to manage, such as infections caused by bacteria resistant to multiple antibiotics [22,25,28,34,41]. In 85% of the cases, the therapy was successful through the administration of either a single phage or a phage cocktail. One of the primary factors responsible for this success is the use of specific bacteriophages for each bacterial strain. Although 100% of the studies reported having conducted in vitro studies of bacteriophage activity at the beginning of the therapy, in 15% of the cases, the infection had not been resolved at the end of the therapy. Some authors attributed these failures to defective modes of administration [38], a low phage concentration [23], or co-infection with other species of bacterial strains [46], rather than to a low effectiveness of the therapy.

One of the main concerns with PT is the development of phage-resistant strains [19,25]. An in vitro study by Oechslin et al. found that the frequency of spontaneous phage resistance mutations in a susceptible strain of *P. aeruginosa* was around 10^–7^. Additionally, the same authors concluded that, although some mutations confer resistance to phages, these mutations can decrease bacterial fitness when compared to normal growth under less strict in vitro conditions in animals [47]. This could explain the success of PT in some patients despite the presence of phage-resistant bacteria [23,26], which would prove that a patient’s immune system plays an important role in the success of PT in the short, medium, and long terms.

Although PT is presented as an emerging strategy to fight multidrug-resistant bacterial infections, evidence suggests that coadministration of phages and antibiotics could be a more effective strategy for the elimination of infection [19]. In cases where combined therapy was administered, patients recovered from the infection with 100% success, whereas 88% of those patients who received PT alone recovered. This result can be explained as follows: combination therapy may decrease the selection of phage-resistant bacteria [47], in addition to increasing the sensitivity to antibiotics [25,47,48]. Additionally, antibiotics could act as adjuvants that minimize the appearance of other pathogens in immunocompromised patients.

Owing to serious public health threats posed by health care-associated infections (HAIs), such as those caused by *E. coli*, *S. aureus*, *K. pneumoniae*, coagulase-negative *Staphylococcus*, and *Enterococcus faecalis* [49], and given the call made by the WHO for a search for new alternatives in the treatment of multidrug-resistant bacteria, especially those belonging to the ESKAPE group [7], it is not surprising that infections caused by these pathogenic bacteria were more frequently treated with PT (50.8%). A single phage or a cocktail of phages proved effective in the treatment of these infections regardless of their sensitivity or resistance to antibiotics. However, phage cocktails were more widely used (84%) to increase bacterial sensitivity to therapy.

In all studies, the administration of PT was considered safe for patients. The age of the treated patients ranged between 2 and 88 years, with an average of 55 years. Duplessis et al. (2017) reported the case of a 2-year-old patient with DiGeorge syndrome and a series of comorbidities that were difficult to manage, in addition to recalcitrant bacteremia/sepsis owing to multidrug-resistant *P. aeruginosa*. Phage therapy was initiated in this patient to treat the infection. Although sterile blood cultures were obtained for *P. aeruginosa* in the short term, the treatment had to be interrupted owing to decompensation of the infant caused by anaphylaxis. A refractory infection led to the reinitiation of PT, which once again proved effective, resulting in sterile blood cultures 24 h after therapy onset. Unfortunately, the patient’s decompensation and worsening prognosis owing to severe arrhythmias as well as cardiogenic and septic shocks (caused by the same bacteria), possibly owing to a history of influenza infection and end-stage heart failure, among others (although none of these effects were related to the administration of PT), led relatives to withdraw care, and the child died soon afterwards [36].

Jault et al. (2019) reported the death of an adult man over 70 years of age who was recruited to participate in the controlled clinical trial PhagoBurn, once the inclusion and exclusion criteria were met. The patient received PT topically for the treatment of *P. aeruginosa*, and his death occurred sometime after completing the treatment regimen, although no association with PT was found [23].

The potential development of adverse events associated with the concentration of endotoxin residues, i.e., the amount of bacterial endotoxin lipopolysaccharide released during phage-mediated lysis [19], has apparently been discarded based on reports that the maximum dose is below 5 units of endotoxin per kg of body weight per hour (required by international regulators), and this is reflected by the absence of serious adverse events during the administration of PT in all the studies. This confirms the safety of PT for the treatment of bacterial infections.

Additionally, bacteriophages offer new advantages, namely, their high specificity toward the host cell, which would reduce the damage to the normal microbiome of the patient and decrease colonization by other pathogens in the absence of in vivo drug interactions, bactericidal activity, minimal variability in pharmacokinetics and pharmacodynamics, agnostic bacterial targeting irrespective of the antibacterial susceptibility profile, minimal environmental footprints, and potential inducement of susceptible bacterial profiles [36]. Moreover, their ability to alter the formation of biofilms enhances the action of antibiotics in combination therapies [21]. In addition, free genetic exchange of phages rarely occurs [3,17,18]. Thus, as the frequency of therapy increases, the treatment doses and periods needed to achieve an optimal effect would be expected to decrease [17,18]. However, one of the main challenges that must be faced for the definitive implementation of phage therapy is the unification of criteria and standardized procedures that allow dealing with infections caused by multidrug-resistant bacteria. This is because the few studies that have been reported present a great variability of results because they not only depend on the multidrug-resistant bacteria, but also on the type and place of the infection that they are causing. This in turn affects the dosage and application of combination therapies, which makes comparison between studies difficult. All this highlights the need to generate guidelines with unified criteria to determine the validity of phage therapy.

## 5. Conclusions

Phage therapy has proven to be effective and safe for the treatment of infectious diseases caused by various bacterial species, including multidrug-resistant strains. This effectiveness has been demonstrated both through local and systemic administration, with no associated serious adverse events. However, much of the success of PT depends on the selection of specific bacteriophages for each bacterial species and strain, owing to their specificity toward the host cell, which is why extensive libraries of phages are required to personalize treatments. On the other hand, coadministration of antibiotics as adjuvants is recommended as it prevents infections by non-specific bacteria and helps eliminate bacteria already sensitized by the phages.

## Figures and Tables

**Figure 1 ijms-23-04577-f001:**
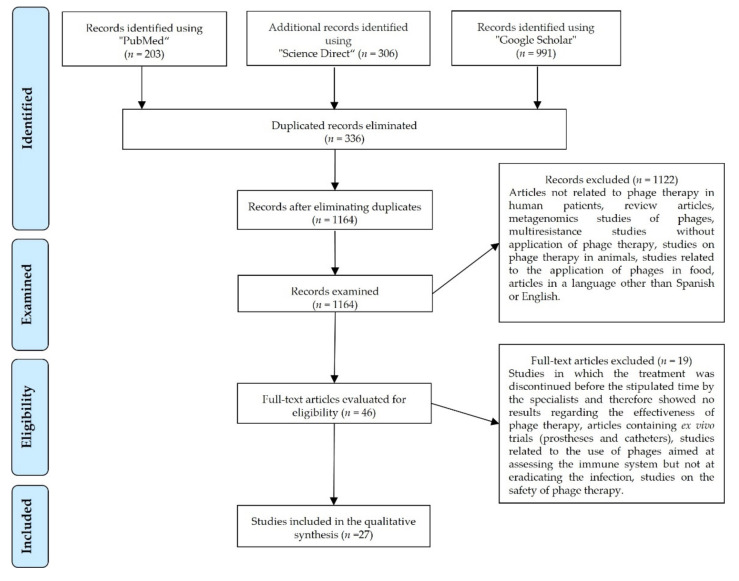
PRISMA flowchart illustrating the search and selection of the articles.

**Figure 2 ijms-23-04577-f002:**
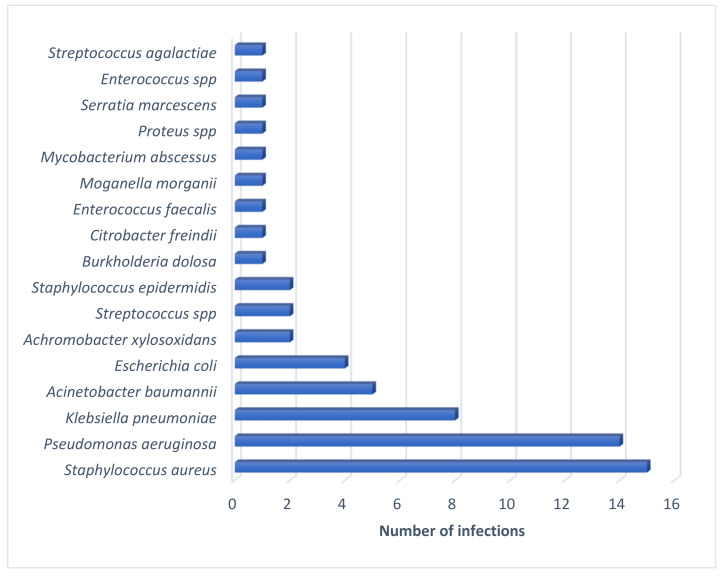
Number of infections caused by each of the multidrug-resistant bacteria mentioned in the articles.

**Table 2 ijms-23-04577-t002:** Characteristics of the bacteriophages used for the treatment of infections.

Author(s)/ Reference	Phage/Cocktail Identification	Number of Phages in the Cocktail	Phage Order or Family	Route of Administration	Concentration Per Unit Dose/Duration of Treatment	Antibiotics
Ooi, M.L. et al. [37]	Cocktail AB-SA01	3	*Myoviridae*	Nasal irrigation	3 × 10^8^–10^9^ PFU, for 14 days	NA
Schooley, R.T. et al. [25]	AB-Navy1-AB-Navy4-AB-Navy71-AB-Navy97-AbTP3Φ1-AC4-C1P12-C2P21-C2P24, cocktails ΦPC-ΦIV-ΦIVB	8	*Myoviridae-Podoviridae* (AbTP3Φ1)	Intravenous-Intracavitary	5 × 10^9^ PFU, for 2 to 16 weeks	Meropenem, fluconazole, and minocycline
Jault, P. et al. [23]	Cocktail PP1131	12	Not reported	Topical	1 × 10^6^ PFU, for 7 days	NA
Law, N. et al. [34]	Cocktail ABPA01	4	Not reported	Intravenous	4 × 10^9^ PFU, for 8 weeks	Ciprofloxacin, piperacillin–tazobactam, doripenem
Aslam, S. et al. [19]	Cocktails AB-PA01, AB-PA01-m1, Navy 1, Navy 2	4, 5, 2, 3	Not reported	Intravenous	Between 5.3 × 10^6^ PFU and 5 × 10^9^ PFU for 3 to 5 weeks	Piperacillin-tazobactam and colistin
	Cocktail AB-PA01	4				Piperacillin-tazobactam, tobramycin, and inhaled colistin
	BdPF16phi4281	1				Meropenem, ceftazidime-avibactam, minocycline, and tobramycin
Nir-Paz, R. et al. [35]	ФAbKT21phi3 and ФKpKT21phi1	2	Not reported	Intravenous	5 × 10^7^ PFU for 5 days	Colistin and meropenem
Tkhilaisshvili, T. et al. [33]	Local phage adjuvants	Not reported	Not reported	Intravenous (local administration system)	10^8^ PFU for 5 days	Colistin, meropenem, and ceftazidime
Corbellino, M. et al. [28]	vB_KpnM_GF	Not reported	Not reported	Oral and intrarectal	Did not report concentration, for 3 weeks	NA
Duplessis, C. et al. [36]	Phage cocktail	2	Not reported	Intravenous	3.5 × 10^5^ PFU, did not specify duration of treatment	NA
Qin, J. et al. [24]	Phage cocktails Ф902, ФJD905, ФJD907, ФJD908, JD910	5	*Podoviridae*, *Myoviridae* (ФJD905)	Irrigations	5 × 10^8^ PFU for 4 weeks	piperacillin– tazobactam, imipenem, amikacin
Tan, X. et al. [26]	Unique Ab_SZ3 phage	1	*Siphoviridae*	Airway (vibrating mesh nebulizer)	Increasing concentrations: 5 × 10^6^ on day 1 to 2.5 × 10^7^ PFU, day 2 with 10^8^ PFU, day 4 with 10^9^ PFU, and day 13 with 5 × 10^10^ PFU for 16 days	Tigecycline and polymyxin E
Hoyle, N. et al. [29]	Phage cocktail	2	*Siphoviridae*	Inhalation (compressor nebulizer)	3 × 10^8^ PFU for 20 days	Piperacillin- tazobactam
Leiter, L. et al. [38]	PYO cocktail	Not reported	Not reported	Intravesical instillation	10^4^ to 10^5^ PFU for 7 days	NA
Cano, E.J. et al. [30]	Phage KpJH46Φ2	1	Not reported	Peripheral intravenous	40 doses of 6.3 × 10^10^ PFU/mL for 30 min plus minocycline	Minocycline
Gupta, P. et al. [44]	Phage cocktail	3	Not reported	Topical	0.1 mL/cm^2^ (a concentration of 10^9^ PFU/mL from 9 to 13 days)	NA
Lebeaux, D. et al. [45]	JW Delta, JWT, and 2-1 cocktails (APC 1.1)	3	Not reported	Inhalation (vibrating mesh nebulizer)	4 × 10^10^ PFU (3 nebulizations/day of 5 mL of solution). Did not specify the duration of treatment.	Imipenem
	Cocktail 2, plus phage JW alpha (APC 2.1)	4	Not reported	Administered to the lungs using a fibroscope and through nebulization	5 × 10^9^ PFU/mL. Did not specify the duration of treatment.	NA
Nadareishvili, L. et al. [40]	Staphylococcal bacteriophage, PYO bacteriophage, and SES bacteriophage	3	Not reported	Topical and oral	Did not specify the concentration; 10 mL once a day for 20 days and then for 2 weeks	NA
	Staphylococcus bacteriophage and Intesti bacteriophage	2	Not reported	Topical and oral	Did not specify the concentration; 10 mL once a day for 20 days initially and then for 15 days	NA
	Staphylococcus bacteriophage and Intesti bacteriophage	2	Not reported	Topical and oral	Did not specify the concentration; 10 mL once a day for 20 days initially and then for 20 days	NA
	Pyo bacteriophage and Intesti bacteriophage	2	Not reported	Topical and oral	Did not specify the concentration; 10 mL once a day for 20 days initially and then for 20 days. Solcoseryl (epithelial regenerator) was also used.	NA
Patel, D.R. et al. [46]	Phage cocktail	Not reported	Not reported	Topical	Did not specify the concentration or the duration of treatment; 500 μL/cm^2^, 5–7 applications	NA
Fadlallah, A. et al. [31]	Phage SATA-8505 (ATCC PTA-9476)	1	Not reported	Topical (eye drops and nasal spray) and general (intravenous)	Did not specify the concentration, for 4 weeks	NA
Jennes, S. et al. [41]	BFC1 Cocktail	2	Not reported	50 microliters administered through intravenous infusion for 6 h over 10 days. Wounds were irrigated with 50 mL of BFC1 every 8 h for 10 days.	Did not specify the concentration, for 10 days	NA
Kuipers, S. et al. [42]	Anti-*Klebsiella pneumoniae* phages	Not reported	Not reported	Oral and bladder irrigation/ intravesical	Did not report the concentration, for 8 weeks	NA
Onsea, S. et al. [43]	COCKTAIL BCF1	3	Not reported	Topical and in marrow	10^7^ PFU/mL for 7 days	Vancomycin, rifampicin, moxifloxacin
	COCKTAIL BCF1	3	Not reported	Topical	10^7^ PFU/mL for 10 days	Vancomycin, colistin, fosfomycin
	COCKTAIL BCF1	3	Not reported	Topical	10^7^ PFU/mL for 9 days	Vancomycin, clindamycin, moxifloxacin
	PYO phage	Not reported	Not reported	Topical	10^7^ PFU/mL for 7 days	Amoxicillin
Wu, N. et al. [27]	Phage 1Ф (ФAb124)	1	*Podoviridae*	Respiratory by inhalation	10^8^ PFU/mL for 20 min, two doses with 1 h interval	NA
	Cocktail 2Ф (ФAb124 and ФAb121)	2	*Podoviridae* and *Myoviridae*	Respiratory by inhalation	1 × 10^10^ PFU, did not specify concentration	NA
	Cocktail 2Ф (ФAb124 and ФAb121)	2	*Podoviridae* and *Myoviridae*	Respiratory by inhalation and topical with a wet compress	1 × 10^10^ PFU, did not specify concentration	NA
	Cocktail 2Ф (ФAb124 and ФAb121)	2	*Podoviridae* and *Myoviridae*	Respiratory by inhalation	1 × 10^10^ PFU, did not specify concentration	NA
	Cocktail 2Ф (ФAb124 and ФAb121)	2	*Podoviridae* and *Myoviridae*	Respiratory by inhalation	1 × 10^10^ PFU, did not specify concentration	NA
Bao, J. et al. [22]	Cocktail I (SZ-1, SZ-2, SZ-3, SZ-6) against the CX7224 strain	5	Not reported	Bladder irrigation	5 × 10^8^ PFU per mL of each phage for 5 days	NA
	Cocktail II (Kp165, Kp166, Kp167, Kp158, and Kp169) for the CX8070 strain	5	Not reported	Bladder irrigation	Did not specify the concentration, for 5 days	NA
	Cocktail III (Kp152, Kp154, Kp155, Kp164, Kp6377, and HD001)	6	Not reported	Bladder irrigation	SMZ-TMP administered orally (800–160 mg) twice daily, plus the cocktail for 5 days	NA
Rose, T. et al. [39]	BFC-1 cocktail (*P*. *aeruginosa* phage 14/1, *S*. *aureus* phages ISP and PNM)	3	*Podoviridae* and *Myoviridae*	Topical	10^9^ PFU/mL, did not specify the duration of treatment	Ceftazidime or meropenem in combination with amikacin. Vancomycin or linezolid.
Rubalskii, E. et al. [32]	*Staphylococcus phage* CH1, Enterococcus phage Enf1, Pseudomonas phage PA5, Pseudomonas phage PA10	4	*Myoviridae*, *Siphoviridae*, *Myoviridae*, *Myoviridae*	Local	1 × 10^8^ PFU/ml	Cefepime, daptomycin, linezolid, tobramycin
	*Klebsiella phage* KPV811, Klebsiella phage KPV15	2	*Podoviridae*, *Myoviridae*	Respiratory by inhalation, via nasogastric tube	1 × 10^8^ PFU/ml	Ceftazidime, linezolid, avibactam, colistin, meropenem, cotrimoxazole, tobramycin
	*Staphylococcus phage* CH1	1	*Myoviridae*	local application via drainage	1 × 10^9^ PFU/ml	Rifampicin, flucloxacillin
	*Staphylococcus phage* CH1	1	*Myoviridae*	local application via drainage	1 × 10^9^ PFU/ml	Daptomycin
	Staphylococcus phage Sa30, *Staphylococcus* phage CH1, Staphylococcus phage SCH1, *Staphylococcus* phage SCH111	4	*Myoviridae*, *Myoviridae*, *Podoviridae*, *Podoviridae*	Local application via drainage, intranasal	1 ×10^9^ PFU/ml	Daptomycin
	*Staphylococcus* phage Sa30	1	*Myoviridae*	Locally, intraoperatively mixed with fibrin glue	4 × 10^10^ PFU/ml	Sultamicillin
	*Escherichia phage* ECD7, *Escherichia phage* V18	2	*Myoviridae*	Locally, intraoperatively mixed with fibrin glue	4 × 10^10^ PFU/ml	Clindamycin
	*Pseudomonas phage* PA5, *Pseudomonas phage* PA10	2	*Myoviridae*	Locally, intraoperatively mixed with fibrin glue	4 ×10^10^ PFU/ml	Colistin, ceftazidime, avibactam

## Data Availability

The data used to support the findings of this study are included within the article and are available from the corresponding author upon request.

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
