# Peer review of "Phage Therapy in the Era of Multidrug Resistance in Bacteria: A Systematic Review"

_ijms, 2022, doi:10.3390/ijms23094577_

Round 1

Reviewer 1 Report

In this paper the authors summarized the data on phage therapy related to human for last 10 years. The text is well written and organized, but could be more detailed about the advantage of phage therapy compared to antimicrobials and the challenges that will be faced.

Major comments:
1.     From line (253~259) author mentioned very few words regarding the advantages of phage therapy, it can be possible to write more advantages specifically on phage therapy in discussion.
2.     Phage therapy still has some challenges, the author can mention in the discussion part. 

Minor comments:
1.     Line 37 ~ 39
Infections caused by multidrug-resistant bacteria have currently become one of the leading causes of morbidity and mortality worldwide, with approximately 700,000 deaths each year This data seems not found in the given references 2,7,8
2.     Line 44
According to the Organization for Economic Cooperation and 43 Development (OECD), 2.4 million people could die from now to 2050 if current bacterial 44 resistance rates continue; this data probably applicable for Europe, North America and Australia that didn’t mention in manuscript. (Link: https://www.who.int/docs/default-source/documents/no-time-to-wait-securing-the-future-from-drug-resistant-infections-en.pdf)
3.     Line 191
Should be a  ]  at the end of all references number
4.     Line 184
The title 3. Discussion, I think it will be 4. Discussion  

Reviewer 2 Report

The reviewed paper presents the summarization of properly described clinical studies using phages to treat antibiotic-resistant pathogenic bacteria. The criteria for choosing the publication for review are properly presented, and the results are conveniently tabulated. Observed success rate is reported as 85% which is substantial to consider this approach for medication. 

On the reviewer's opinion, the paper is rather descriptive and inconclusive; However this is not a fault of the Authors, but can be explained by inconsistent presentation of the results in the original papers. Very few can be said about the choice, dosage and application strategy of therapeutic phages, and the results of the treatment of highly variable pathogens and types of infection are hard to compare. 

Nevertheless, this manuscript is worth a publication to outline the need for unified criteria for assessment of the validity of phage therapy. Probably, this issue is worth to underline in the "Discussion". 

Minor comments:

line 53 - because OF their capacity

line 58 - natural MICROflora

Figure 1 - middle right panel "phage therapy" instead of "phagotherapy"

line 191 - reference bracket

The reference list is presented very poorly. Authors "P.F., G.V. and L.TK" are not informative. This usually happens due to reference manager errors, and should be verified manually.

Reviewer 3 Report

Carlos Aranga et al., present a systematic review concerning phage therapy in the era of multidrug resistance in bacteria. Although, the topic is very interesting I have a few concerns.

Please, read my comments below.

References:

  • I think that citations such as: “D.A. Spricigo, “Thesis Doctoral” shouldn’t be included in the references. The thesis is not precisely cited and it is not in accordance with the standards of scientific writing.
  • The references are not presented uniformly. Sometimes only authors initials are included, sometimes the whole surnames. Please, always list the author's surname.
  • The search strategy was defined by phrases: “phage therapy” and “multidrug resistant”.

Therefore, some relevant publications that concern phage therapy in clinical setting and “pan-resistant bacteria” did not drag the attention of the authors. The publications such as:

Rubalskii et al., Bacteriophage therapy for critical infections related to cardiothoracic surgery; Antibiotics 2020, 9(5), 232; https://doi.org/10.3390/antibiotics9050232

Lebeaux et al., A Case of Phage Therapy against Pandrug-Resistant Achromobacter xylosoxidans in a 12-Year-Old Lung-Transplanted Cystic Fibrosis Patient; Viruses. 2021 Jan 5;13(1):60. doi: 10.3390/v13010060

Eskanazi et al., Combination of pre-adapted bacteriophage therapy and antibiotics for treatment of fracture-related infection due to pandrug-resistant Klebsiella pneumoniae; Nat Commun 13, 302 (2022). https://doi.org/10.1038/s41467-021-27656-z

should be included in this review. I think that the search may be redefined with use of: “pan-drug resistant”.  

  • Although, in the Discussion section and in Conclusions the authors write that the combined therapy: bacteriophages + antibiotics may be more efficient in eradication of bacteria than phage therapy alone, the authors do not state what kind of antibiotics were administered to patients, at what doses and whether they were combined with single phages or phage cocktails. Because, the authors write that combined therapy was more successful than PT alone, this important information must be included in the manuscript, either in the Table or in the text.

Round 2

Reviewer 3 Report

The authors satisfactory addressed all my queries. Therefore, the manuscript in the present form is suitable for publication in IJMS.